# Associations of Serum Vitamin A and E Concentrations with Pulmonary Function Parameters and Chronic Obstructive Pulmonary Disease

**DOI:** 10.3390/nu16183197

**Published:** 2024-09-21

**Authors:** Wonjun Noh, Inkyung Baik

**Affiliations:** Department of Foods and Nutrition, College of Science and Technology, Kookmin University, Seoul 02707, Republic of Korea; nwj1228@kookmin.ac.kr

**Keywords:** vitamin A, vitamin E, pulmonary function tests, chronic obstructive pulmonary disease

## Abstract

Background/Objectives: Oxidative stress, an imbalance between oxidants and antioxidants, is known to affect pulmonary function (PF), thereby leading to the development of chronic obstructive pulmonary disease (COPD). However, data on the associations of serum vitamin A and E concentrations with PF parameters and COPD are inconsistent. The present cross-sectional study aimed to investigate these associations, considering inflammatory status. Participants/Methods: This study included 2005 male and female adults aged ≥40 years who had participated in a population-based national survey. Spirometry without a bronchodilator was conducted to yield PF parameters, such as forced expiratory volume in one second (FEV1), forced vital capacity (FVC), and the FEV1/FVC ratio, which were used to define COPD. Serum vitamin A (retinol) and E (α-tocopherol) concentrations were assayed. Multivariable regression analysis was performed after adjusting for potential confounding variables. Results: Serum vitamin A concentration was positively associated with FEV1 (*p* for trend < 0.01) among all participants. In addition, the odds ratio (95% confidence interval) of the highest serum vitamin A concentration tertile for the prevalence of COPD, which was defined by the FEV1/FVC ratio < 0.7, was 0.53 (0.31, 0.90) compared with that of the lowest tertile (*p* for trend < 0.05). Analysis stratified by a cutoff point of 1 mg/L serum high-sensitivity C-reactive protein (hs-CRP) revealed that such associations with FEV1 and COPD prevalence were stronger in participants with lower hs-CRP levels (*p* for trend < 0.05). In contrast, serum vitamin E concentration was associated with neither PF parameters nor COPD prevalence. Conclusions: These findings suggest that serum vitamin A concentration may be important in preventing the progressive decline in PF parameters that results in COPD. Further epidemiological investigations are warranted to evaluate the causal associations of antioxidant vitamin status with PF parameters and COPD.

## 1. Introduction

Chronic obstructive pulmonary disease (COPD) is a common pulmonary disease characterized by progressive airflow limitation, leading to breathing difficulties. The global prevalence of COPD was estimated to be approximately 10% among adults aged 30–79 years in 2019 [1]. Similarly, the prevalence of COPD was reportedly 12.9% among Korean adults aged ≥40 years [2]. In older adults aged ≥70 years, both reports estimated a notable prevalence more than double the overall prevalence [1,2]. As the older population increases, COPD is emerging as a significant and escalating public health problem worldwide.

Smoking, air pollution, and occupational exposure to dust or smoke are well-known major factors for the increased risk of developing COPD [1,3]. These risk factors presumably induce an imbalance between oxidants and antioxidants, thereby aggravating oxidative stress, which is ostensibly integral to the pathogenesis of COPD [4]. In contrast, endogenous and exogenous (dietary) antioxidants exert protective effects against oxidative stress; therefore, antioxidant status is further expected to be positively associated with pulmonary function (PF) and inversely associated with the risk of COPD [4]. In fact, a meta-analysis analyzing accumulated data on the association of dietary antioxidants with COPD symptoms and outcomes observed a significantly inverse association for vitamins C and E, but no association for vitamin A [5]. Epidemiological data on the association of the blood concentrations of these vitamins, which are objective markers of vitamin status, with PF and COPD prevalence are inconsistent [6,7,8,9,10,11,12,13]. Discrepancies in these findings may partly emanate from a lack of the following considerations: features of impaired PF (obstructive or non-obstructive patterns), inflammatory status, dietary supplementation, and the adjustment of total energy intake, which appear to be a potential confounder because COPD is accompanied by an energy imbalance owing to a reduced dietary intake and an increased resting energy expenditure [14].

The present study focused on serum vitamin A (retinol) and E (α-tocopherol) concentrations because these two antioxidant vitamins were exclusively analyzed in a national survey, whose data we utilized, to investigate their associations with PF parameters and COPD based on the evaluation of obstructive and non-obstructive patterns. In this analysis, the concentration of serum high-sensitivity C-reactive protein (hs-CRP), which is a marker of systemic inflammation, was considered as a confounder and an effect modifier, and dietary supplementation and total energy intake as confounding variables.

## 2. Participants and Methods

### 2.1. Participants

The study participants were male and female adults aged ≥40 years from the Korea National Health and Nutrition Examination Survey (KNHANES) conducted in the 7th survey period (2016–2018). The KNHANES, a population-based, cross-sectional survey that employs a complex stratified sampling design, is conducted by the Korea Disease Control and Prevention Agency (https://knhanes.kdca.go.kr/knhanes/ (accessed on 20 August 2024)). KNHANES participants comprised noninstitutionalized Korean citizens residing in South Korea who had signed an informed consent form [15]. In a total of 21,273 participants from the 7th KNHANES (*n* = 7042 in 2016, *n* = 7167 in 2017, and *n* = 7064 in 2018), 12,225 were eligible adults aged ≥ 40 years. After excluding individuals who had not undergone PF testing (*n* = 3830) or had not reported ≥ 8 h of fasting before blood sample collection for serum vitamin A and E concentration assays (*n* = 5597), those who had been diagnosed with chronic diseases (*n* = 422), such as COPD (*n* = 21), asthma (*n* = 70), pulmonary tuberculosis (*n* = 102), lung cancer (*n* = 4), stroke (*n* = 38), cardiovascular disease (*n* = 59), renal failure (*n* = 4), liver cirrhosis (*n* = 11), and cancer (*n* = 113), by a physician were excluded to eliminate the effects of chronic diseases on serum vitamin concentrations. In addition, individuals who had been found to have non-obstructive lung disease (restrictive pattern PF) based on the PF test (*n* = 294) or had not reported information (missing data) on smoking, alcohol consumption, or other confounding variables (*n* = 77) were excluded. Finally, data for 2005 participants (829 men and 1176 women) were analyzed. The present study was approved by the Institutional Review Board of Kookmin University (approval number: KMU-202102-HR-260).

### 2.2. Definition of Pulmonary Function Parameters

In the KNHANES, only participants aged ≥40 years underwent PF testing; the procedure and quality control has been described in a report [16]. Briefly, trained personnel conducted PF testing using a spirometer (a dry rolling seal spirometer was used until June 2016, after which a Vyntus™ SPIRO spirometer [CareFusion, San Diego, CA, USA] was employed) without a bronchodilator (post-bronchodilator testing was not performed in the KNHANES) and generated PF parameters, including forced expiratory volume in one second (FEV1), forced vital capacity (FVC), and the FEV1/FVC ratio, which were utilized as analysis outcomes. In the KNHANES, three PF classifications were defined based on the test results as follows: (1) normal PF (FEV1/FVC ≥ 0.7 and FVC ≥ 80% of the predicted value), (2) a restrictive pattern of impaired PF (FEV1/FVC ≥ 0.7 and FVC < 80% of the predicted value), and (3) an obstructive pattern of impaired PF (FEV1/FVC < 0.7) [17]. Among the participants who had been diagnosed with impaired PF, those with an obstructive pattern were exclusively included and considered as COPD cases [18].

### 2.3. Serum Vitamin A and E and High-Sensitivity C-Reactive Protein Assay

Serum concentrations of vitamins A (retinol) and E (α-tocopherol) were assayed using the Agilent 1200 high-pressure liquid chromatography (HPLC) system (Agilent, Santa Clara, CA, USA) and HPLC reagent kits (Chromsystems, Gräfelfing, Germany) and hs-CRP levels using the Cobas Analyzer (Roche Diagnostics, Indianapolis, IN, USA) by a commercial laboratory that performed quality control tests [19].

### 2.4. Potential Confounding Variables and an Effect Modifier

Data on demographic and health-related characteristics including age, sex, residential district, educational level, household income level, body mass index (BMI) calculated by dividing body weight (kg) by the square of the height (m), smoking status, alcohol consumption, coffee consumption, regular physical activity (defined based on the performance of medium-intensity physical activity for ≥2 h 30 min per week or that of high-intensity physical activity for ≥1 h 15 min), total calorie intake, and dietary supplementation (vitamins, minerals, and functional foods) were collected from questionnaire-based data, and considered to be potential confounding variables in the analysis. In addition, serum hs-CRP concentration was used as a confounder and an effect modifier in the analysis because it was reportedly associated with antioxidant status and COPD [20,21].

### 2.5. Statistical Analysis

According to the serum vitamin A and E concentration tertile groups, descriptive statistics were obtained with consideration of sampling weight and presented as the mean ± standard error (SE) or percentage. Statistical differences between groups were evaluated using analysis of variance and the chi-square test for continuous and categorical variables, respectively. To analyze the associations between serum vitamin concentrations and PF parameters, linear regression analysis was performed with consideration of sampling weight and potential confounding variables to obtain regression coefficient estimates and their SEs. To satisfy the normality assumption, each PF parameter was squared and fitted as a dependent variable in the model based on the variable transformation results. In the multivariable models, age, BMI, total calorie intake, and hs-CRP level were fitted as continuous variables, while sex, residential district (rural, urban), educational level (middle school or lower, high school or higher), household income level (low and lower-middle, upper-middle and high), smoking status (abstainer, current smoker [two categories: <20, ≥20 pack-years]), alcohol consumption (abstainer, drinker), coffee consumption (non-consumer, consumer), regular physical activity (no, yes), and dietary supplementation (no, yes) were fitted as categorical variables. Additionally, multivariable association analysis, stratified by serum hs-CRP level (≤1, >1 mg/L), was conducted. To analyze the association between serum vitamin concentrations and COPD prevalence, logistic regression analysis was performed with consideration of sampling weight and previously mentioned potential confounding variables to obtain odds ratios (ORs) and 95% confidence intervals (CIs).

Statistical analyses were performed using SAS v.9.4 software (SAS Institute Inc., Cary, NC, USA). Statistical significance was set at *p* < 0.05.

## 3. Results

### 3.1. Characteristics of the Study Participants

Among 2005 study participants, 224 (weighted prevalence: 9.9%) were observed to have impaired PF, especially obstructive pattern PF. The study participants were compared among serum vitamin A and E concentration tertiles (Table 1). Participants in the lowest serum vitamin A tertile were more likely to be women, leaner, current non-smokers, and current non-drinkers of alcohol, as well as to consume fewer calories and exhibit higher serum hs-CRP concentrations. However, those in the lowest serum vitamin E tertile were more likely to be men, consume fewer dietary supplements, and exhibit lower serum hs-CRP concentrations.

### 3.2. Associations between Serum Vitamin Concentrations and PF Parameters

Table 2 shows the associations of serum vitamin A and E concentrations with PF parameters. Positive associations of serum vitamin A concentration with FEV1 (*p* for trend < 0.01) and FVC (*p* for trend = 0.061) were observed in the multivariable models. However, no associations between serum vitamin E concentration and PF parameters were observed.

### 3.3. Associations between Serum Vitamin Concentrations and PF Parameters Stratified by Serum hs-CRP Levels

As shown in Table 3, which displays the association results stratified by serum hs-CRP level, serum vitamin A concentration was significantly associated with FEV1 exclusively in participants with hs-CRP levels ≤ 1 mg/L, indicating a low systemic inflammatory state (*p* for trend < 0.05). However, stratified analysis revealed no associations between serum vitamin E concentration and PF parameters.

### 3.4. Associations between Serum Vitamin Concentrations and COPD with Results Stratified by Serum hs-CRP Level

Table 4 shows the associations of serum vitamin A and E concentrations with the prevalence of spirometry-defined COPD. Similar to the results depicted in Table 2, serum vitamin A concentration was significantly associated with a lower COPD prevalence across all participants in the multivariable model (*p* for trend < 0.05). Compared with the bottom serum vitamin A concentration tertile group, the top tertile group yielded a lower OR (0.53 (95% CI: 0.31, 0.90)) for COPD. According to the association results stratified by serum hs-CRP level, participants with a low systemic inflammatory state of this marker (≤1 mg/L) exclusively exhibited a relatively low COPD prevalence; their ORs (95% CI) were 0.55 (0.34, 0.91) and 0.51 (0.30, 0.88) for the second and third tertile groups, respectively. However, such a low prevalence was not observed among those with serum hs-CRP levels > 1 mg/L. Meanwhile, serum vitamin E concentration and COPD prevalence were not associated.

## 4. Discussion

This cross-sectional study analyzed nationwide survey data to investigate the associations of serum vitamin A and E concentrations with PF parameters and COPD prevalence in middle-aged and older adults. Overall, serum vitamin A concentration exhibited significantly positive associations with FEV1 and a low prevalence of COPD. In particular, analyses stratified by serum hs-CRP level revealed that participants with serum hs-CRP levels ≤1 mg/L, which indicate a low inflammatory state and were observed in 76% of the study participants, displayed such associations. However, serum vitamin E concentration exhibited no association with PF parameters and COPD prevalence.

Oxidative stress, which is caused by the production of reactive oxygen species (ROS), thus overwhelming antioxidant capacity, has been proposed as a potential mechanism in the pathogenesis of COPD. ROS, including the superoxide anion, hydrogen peroxide, the hydroxyl radical, and single oxygen, are produced endogenously via metabolic reactions; furthermore, they are generated by exogenous sources such as cigarette smoke and air pollutants [4]. In pulmonary tissue, increased levels of oxidative stress are accompanied by inflammation, damage epithelial cells, and induce extracellular matrix degradation, leading to irreversible injury and the development of fibrosis [22]. However, because endogenous (e.g., glutathione, coenzyme Q10, superoxide dismutase, catalase, and glutathione peroxidase) and dietary exogenous antioxidants, including vitamins A and E, can scavenge ROS, thereby diminishing oxidative stress, they presumably play a protective role in preventing PF decline and COPD development [5].

Epidemiological data on the association of serum vitamin A and E concentrations with PF remain inconsistent [6,7,8,9,10]. Certain population-based, cross-sectional studies have observed a significant association of PF parameters with serum retinol [8] and β-carotene concentrations [7,8]; nevertheless, others have not [6,9]. Regarding investigations on serum vitamin E, particularly α-tocopherol, no associations [6,10], positive associations [7,8], and an inverse association [9] with PF parameters have been reported. Concerning the association with COPD, conflicting data have been generated by case–control studies, which performed PF testing to diagnose COPD [11,12]. A national survey involving American adults found that those who had reported a diagnosis of COPD were likely to have lower serum α-carotene, β-carotene, and α-tocopherol concentrations [13]. Data from large sample-size studies on such associations with COPD, especially cases diagnosed via PF testing, remain limited.

The current study is based on the same national survey (the KNHANES, conducted during the 2016–2018 period) data used in an earlier study [9] that exclusively included older adults; nonetheless, our study extended the scope of the study participants to adults aged ≥ 40 years. In addition, a broader range of confounding variables, including total calorie intake, dietary supplementation, and serum hs-CRP concentration, was considered in the multivariable models. The aforementioned previous study observed no association with serum vitamin A (retinol) concentration and an inverse association with serum vitamin E (α-tocopherol) concentration when PF parameters were fitted as dependent variables [9]. We speculated that the sample size limited to older adults may reduce statistical power, and that residual confounding from unadjusted factors, such as inadequate dietary intake or dietary supplementation, accompanied by severe PF impairment in older adults, may influence the previous results yielded by Chang et al. [9]. Our findings wherein serum retinol concentrations were associated with PF parameters and COPD prevalence are consistent with those of previous studies [8,11]. The results revealing no association with serum α-tocopherol concentration are consistent with those obtained by Hanson et al. [10], but not with the results of the other studies [7,8,13].

The current study conducted a novel investigation into the associations of serum vitamin A and E concentrations with PF parameters and COPD prevalence according to the serum hs-CRP level. Hs-CRP is reportedly a useful biochemical marker for determining COPD severity [21]. Patients with COPD exhibit increased oxidative stress accompanied by airway inflammation [23]. To mitigate oxidative-stress-induced lung damage, enzymatic and non-enzymatic antioxidant (e.g., vitamins A and E) defense mechanisms in the lungs are requisite to protecting against COPD [4]. Our findings indicate that higher serum vitamin A concentrations may help maintain PF in individuals with lower serum hs-CRP levels, but not among those with an elevated inflammatory state. These findings potentially imply that the antioxidant defense mechanism cannot overcome oxidative lung damage in patients with advanced PF impairment accompanied by a high inflammatory state.

The strengths of this study include the use of spirometry measures for outcomes, the analysis of data from a large-scale population-based survey, and the consideration of an extensive range of potential confounding variables. Notwithstanding, this study has certain limitations. First, serum from a single blood draw was assayed for retinol and α-tocopherol concentrations; moreover, other isoforms of vitamins A and E, especially β-carotene and several other carotenoids, γ-tocopherol, and δ-tocopherol, were not examined. Second, the results of the association for vitamin A with the FEV1/FVC ratio and COPD are inconsistent, possibly because of some outliers of the FEV1/FVC ratio, which might lead to a lack of statistical significance. Third, causal inference was limited by the study’s cross-sectional nature. Fourth, possible residual confounding caused by unmeasured variables cannot be ruled out. Finally, the generalization of this study’s findings is limited because the study participants were exclusively Koreans aged ≥40 years.

## 5. Conclusions

In summary, the current study found serum retinol concentration to be positively associated with PF parameters and a relatively low COPD prevalence when confounding variables, including serum hs-CRP level, total energy intake, and dietary supplementation, were considered. Such associations were exclusively observed in individuals with serum CRP level ≤ 1 mg/L. However, we were unable to observe a significant association with serum α-tocopherol concentration. Further epidemiological studies are warranted to generate data on causal inference regarding the associations of the serum concentrations of vitamin A and E, including their various isoforms, which more reliably reflect antioxidant status in the human body. Meanwhile, maintaining proper vitamin A status is an important public health message for adults at a high risk of COPD.

## Figures and Tables

**Table 1 nutrients-16-03197-t001:** Characteristics of 2005 study participants according to the tertiles of serum vitamin A (retinol) and E (α-tocopherol) concentrations.

Variables		All	Serum Vitamin A Tertiles (T) [Median]			*p* Value	Serum Vitamin E Tertiles (T) [Median]			*p* Value
			1st T [0.37]	2nd T [0.51]	3rd T [0.69]		1st T [10.09]	2nd T [13.54]	3rd T [18.43]	
Age, years		54.0 ± 0.25	53.2 ± 0.45	54.7 ± 0.41	54.3 ± 0.42	0.091	53.5 ± 0.46	53.5 ± 0.46	54.6 ± 0.37	0.067
Men, %		48.5	28.4	49.4	66.2	<0.001	52.8	50.1	42.6	<0.01
Residential district	Urban	85.9	86.0	87.3	84.5		88.4	84.1	85.1	0.097
	Rural	14.1	14.0	12.8	15.6		11.6	16.0	14.9	
Educational level	≤Middle school	26.1	23.7	28.4	26.2	0.233	25.6	22.7	30.2	<0.05
	≥High school	73.9	76.3	71.6	73.8		74.5	77.3	69.8	
Low household income level ^1^, %		35.2	34.9	37.4	33.4	0.375	37.4	32.2	35.8	0.240
Body mass index, kg/m^2^		24.1 ± 0.08	23.8 ± 0.14	24.1 ± 0.13	24.4 ± 0.15	<0.01	23.9 ± 0.16	24.1 ± 0.13	24.3 ± 0.13	0.129
Current smoker, %		20.1	8.9	17.3	33.1	<0.001	21.7	18.0	20.6	0.362
Pack-years of cigarettes ^2^		18.8 ± 0.6	16.6 ± 1.39	18.1 ± 1.04	20.0 ± 0.86	0.767	19.5 ± 1.11	18.4 ± 0.91	18.2 ± 1.02	0.311
Current alcohol drinker, %		57.7	45.8	56.9	69.4	<0.001	58.7	59.2	55.2	0.378
Coffee consumer, %		57.5	54.1	58.9	59.4	0.166	58.8	56.5	57.2	0.759
Regular physical activity ^3^, %		47.0	47.2	47.0	46.7	0.989	44.9	47.8	48.3	0.494
Total energy intake, kcal/day		2042.6 ± 24.34	1904.9 ± 38.29	2029.3 ± 40.6	2181.5 ± 43.73	<0.001	2073.0 ± 41.12	2057.3 ± 41.54	1996.6 ± 39.73	0.154
Dietary supplementation, %		55.5	55.3	55.5	55.5	0.982	47.1	57.2	62.5	<0.001
Serum hs-CRP, mg/L		1.06 ± 0.05	1.33 ± 0.13	0.92 ± 0.05	0.93 ± 0.05	<0.001	0.89 ± 0.06	1.05 ± 0.07	1.25 ± 0.11	<0.001
Impaired pulmonary disease, %		9.94	10.07	9.82	9.93	0.990	11.51	8.14	10.13	0.169

Abbreviations: hs-CRP, high-sensitivity C-reactive protein. Values are presented as the mean ± standard error or %. ^1^ Participants whose household income is lower than median; ^2^ calculated in current smokers; ^3^ defined based on the performance of medium-intensity physical activity for ≥2 h 30 min per week or that of high-intensity physical activity for ≥1 h 15 min.

**Table 2 nutrients-16-03197-t002:** Associations of serum vitamin concentrations with lung function parameters among all participants.

Variables	Estimates ± SEs across Vitamin Concentration Tertiles (T)			*p* for Trend across the Tertiles
	1st T	2nd T	3rd T	
*Across serum vitamin A concentration*				
FEV1				
Age- and sex-adjusted model	Reference	0.0150 ± 0.0241	0.0614 ± 0.0271 *	< 0.05
Multivariable model	Reference	0.0165 ± 0.0238	0.0710 ± 0.0274 **	< 0.01
FVC				
Age- and sex-adjusted model	Reference	−0.0171 ± 0.0267	0.0555 ± 0.0312	0.057
Multivariable model	Reference	−0.0156 ± 0.0265	0.0560 ± 0.0321	0.061
FEV1/FVC				
Age- and sex-adjusted model	Reference	0.0088 ± 0.0056	0.0057 ± 0.0066	0.456
Multivariable model	Reference	0.0089 ± 0.0056	0.0092 ± 0.0067	0.204
*Across serum vitamin E concentration*				
FEV1				
Age- and sex-adjusted model	Reference	0.0607 ± 0.0244 *	0.0006 ± 0.0253	0.823
Multivariable model	Reference	0.0493 ± 0.0243 *	−0.0086 ± 0.0263	0.580
FVC				
Age- and sex-adjusted model	Reference	0.0390 ± 0.0303	−0.0031 ± 0.0276	0.805
Multivariable model	Reference	0.0297 ± 0.0301	−0.0112 ± 0.0286	0.596
FEV1/FVC				
Age- and sex-adjusted model	Reference	0.0113 ± 0.0058	0.0008 ± 0.0059	0.960
Multivariable model	Reference	0.0101 ± 0.0058	0.0001 ± 0.0058	0.843

Abbreviations: FEV1, forced expiratory volume in one second; FVC, forced vital capacity. The multivariate models include age, sex, body mass index, residential district (urban, rural), educational level (≤middle school, ≥high school), household income level (low, high), smoking status (abstainer, current smoker [two categories: <20, ≥20 pack-years]), alcohol consumption (abstainer, current drinker), coffee consumption (non-consumer, consumer), regular physical activity (no, yes), dietary supplementation (no, yes), total energy intake, and serum hs-CRP levels. * *p* < 0.05 and ** *p* < 0.01.

**Table 3 nutrients-16-03197-t003:** Multivariable associations between serum vitamin A and E concentrations and pulmonary function parameters according to serum hs-CRP levels.

Variables	Estimates ± SEs across Vitamin Concentration Tertiles (T)			*p* for Trend across the Tertiles
	1st T	2nd T	3rd T	
*Across serum vitamin A concentration*				
hs-CRP ≤ 1 mg/L				
FEV1	Reference	0.0194 ± 0.0271	0.0702 ± 0.0307 *	<0.05
FVC	Reference	−0.0118 ± 0.0312	0.0619 ± 0.0384	0.082
FEV1/FVC	Reference	0.0104 ± 0.0058	0.0081 ± 0.0069	0.470
hs-CRP > 1 mg/L				
FEV1	Reference	0.0313 ± 0.0462	0.0767 ± 0.0601	0.206
FVC	Reference	0.0197 ± 0.0501	0.0595 ± 0.0582	0.302
FEV1/FVC	Reference	0.0021 ± 0.0228	0.0240 ± 0.0165	0.206
*Across serum vitamin E concentration*				
hs-CRP ≤ 1 mg/L				
FEV1	Reference	0.0269 ± 0.0271	0.0136 ± 0.0280	0.701
FVC	Reference	0.0205 ± 0.0348	0.0031 ± 0.0334	0.988
FEV1/FVC	Reference	0.0043 ± 0.0061	0.0019 ± 0.0060	0.592
hs-CRP > 1 mg/L				
FEV1	Reference	0.0812 ± 0.0547	−0.0422 ± 0.0581	0.410
FVC	Reference	0.0509 ± 0.0598	−0.0598 ± 0.0547	0.241
FEV1/FVC	Reference	0.0630 ± 0.0191 **	0.0157 ± 0.0221	0.410

Abbreviations: hs-CRP, high-sensitivity C-reactive protein; FEV1, forced expiratory volume in one second; FVC, forced vital capacity. The multivariate models include age, sex, body mass index, residential district (urban, rural), educational level (≤middle school, ≥high school), household income level (low, high), smoking status (abstainer, current smoker [two categories: <20, ≥20 pack-years]), alcohol consumption (abstainer, current drinker), coffee consumption (non-consumer, consumer), regular physical activity (no, yes), dietary supplementation (no, yes), total energy intake, and serum hs-CRP levels. * *p* < 0.05 and ** *p* < 0.01.

**Table 4 nutrients-16-03197-t004:** Associations between serum vitamin concentrations and prevalence of chronic obstructive pulmonary disease.

Variables	OR (95% CI) across Vitamin Concentration Tertiles (T)			*p* for Trend across the Tertiles
	1st T	2nd T	3rd T	
*Across serum vitamin A concentration*				
Number of cases/noncases	69/614	76/584	79/583	0.055
Age and sex-adjusted model for all	Reference	0.63 (0.40, 1.00)	0.59 (0.37, 0.96) *	<0.05
Multivariable model for all	Reference	0.63 (0.39, 1.02)	0.53 (0.31, 0.90) *	<0.05
Multivariable model for hs-CRP ≤ 1 mg/L	Reference	0.55 (0.34, 0.91) *	0.51 (0.30, 0.88) *	0.450
Multivariable model for hs-CRP > 1 mg/L	Reference	0.87 (0.34, 2.25)	0.71 (0.27, 1.90)	0.055
*Across serum vitamin E concentration*				
Number of cases/noncases	83/585	66/603	75/593	
Age and sex-adjusted model for all	Reference	0.70 (0.46, 1.08)	1.10 (0.73, 1.65)	0.574
Multivariable model for all	Reference	0.79 (0.49, 1.25)	1.22 (0.78, 1.91)	0.324
Multivariable model for hs-CRP ≤ 1 mg/L	Reference	0.91 (0.56, 1.46)	1.10 (0.69, 1.74)	0.482
Multivariable model for hs-CRP > 1 mg/L	Reference	0.67 (0.24, 1.91)	0.76 (0.28, 2.04)	0.605

Abbreviations: OR, odds ratio; CI, confidence interval; hs-CRP, high-sensitivity C-reactive protein. The multivariate models include age, sex, body mass index, residential district (urban, ruraleducational level (≤middle school, ≥high school)), household income level (low, high), smoking status (abstainer, current smoker [two categories: <20, ≥20 pack-years]), alcohol consumption (abstainer, current drinker), coffee consumption (non-consumer, consumer), regular physical activity (no, yes), dietary supplementation (no, yes), total energy intake, and serum hs-CRP levels. * *p* < 0.05.

## Data Availability

All data are publicly available (https://knhanes.kdca.go.kr/knhanes/ (accessed on 20 August 2024)).

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
