# Peer review of "Associations of Serum Vitamin A and E Concentrations with Pulmonary Function Parameters and Chronic Obstructive Pulmonary Disease"

_nutrients, 2024, doi:10.3390/nu16183197_

Round 1

Reviewer 1 Report

Comments and Suggestions for Authors

This is a very interesting study investigating the correlation of vitamins A and E and CRP with lung functions: FEV1, FCV, and their ratio. The authors recruited lots of participants proved that this study has the translational value in south Korean that might help to discovered the potential biomarkers or therapeutic agents in against COPD, or any other lungs disease associated with decreased lung function. Here are some comments for the authors to consider:

1. The authors mentioned that women usually have lower vitamin level, which means the male/ female should be analyzed separated while the authors did not. 

2. The declined lung functions usually accompanied with smoking which is one of the most important factors in causing COPD. performed the correlations to smoking history should be necessary for early diagnosis of COPD development.

3. Please be precise of how to perform the sex-based adjustment, or please provide a table separate the sex and do the correlations. 

Author Response

Manuscript ID: nutrients-3196868

Title: Associations of Serum Vitamin A and E Concentrations with Pulmonary Function Parameters and Chronic Obstructive Pulmonary Disease

We are most grateful with the reviewers’ comments on our manuscript. We have revised the manuscript in accordance with the comments. We list here our response to each comment.

Response to Reviewer 1’s comments

  1. The authors mentioned that women usually have lower vitamin level, which means the male/ female should be analyzed separated while the authors did not.

Response: We appreciate the reviewer's comments and agree on this opinion. In sex-specific analysis, however, significant findings were not observed possibly because of limited statistical power in a reduced sample size for each sex. Although the sex-specific findings could not reach statistical significance, the association trends were similar. In addition, we observed that the sex variable is a potential confounder, but not a significant effect modifier. Thus, as shown in the original result tables, the sex variable was adjusted for in all regression models.

  1. The declined lung functions usually accompanied with smoking which is one of the most important factors in causing COPD. performed the correlations to smoking history should be necessary for early diagnosis of COPD development.

Response: We agree on the reviewer’s opinion. Although we did not present the association results between smoking and COPD in the tables (because our main exposures are serum vitamins), because we observed that smoking is related to COPD, smoking status was adjusted for in the multivariable models.

  1. Please be precise of how to perform the sex-based adjustment, or please provide a table separate the sex and do the correlations.

Response: The sex variable was fitted as a covariate in all regression models for PF parameters and COPD. As mentioned in the first response, because sex-specific associations did not reach statistical significance because of a smaller sample size and the sex variable was not a significant effect modifier, we decided to present the result tables for all sexes. The sex variable is not a continuous variable, we were unable to perform correlation analysis. As requested, however, we provide here the sex-specific association table.

Reviewer 2 Report

Comments and Suggestions for Authors

The authorsreport interesting data from the KNHANES study, on the relationship between PF parameters, in particular FEV1 and FVC, but not FEV1/FVC ratio and Vit A. They report lower COPD cases in those with high Vit A. Furthermore, they have stratified the population based on the level of systemic inflammation, based on hsCRP.

Based on the title of the manuscript , the aim of the study was to assess associations of Vit A and Cit C with pulmonary function paramters and COPD. An important limitation of the current study is the lack of discussion of the pulmonary function outcomes. Important to realize is that the authors report associations with FEV1 anf FVC but not with FEV1/FVC ratio. The authors have to discuss these data from a physiological perspective. At least they have discussed the possible contradiction in their results related to Vit A: no effect on the spirometric definition of COPD but a reduction in COPD cases with higher vit A. What about effects of Vit A on lung size?

Minor comments:

1. What is the normal range of hs CRP in a Korean population > 40 y. The current cut-off of 1 mg/L seems very low even in healthy individuals.

2. In the methods the authors formulated three PF classifications. It seems that the restrictive group is overlooked in the results?

Author Response

Manuscript ID: nutrients-3196868

Title: Associations of Serum Vitamin A and E Concentrations with Pulmonary Function Parameters and Chronic Obstructive Pulmonary Disease

We are most grateful with the reviewers’ comments on our manuscript. We have revised the manuscript in accordance with the comments. We list here our response to each comment.

Response to Reviewer 2’s comments

  1. An important limitation of the current study is the lack of discussion of the pulmonary function outcomes. Important to realize is that the authors report associations with FEV1 and FVC but not with FEV1/FVC ratio. The authors have to discuss these data from a physiological perspective. At least they have discussed the possible contradiction in their results related to Vit A: no effect on the spirometric definition of COPD but a reduction in COPD cases with higher Vit A. What about effects of Vit A on lung size?

Response: We appreciate the reviewer's comments and we are aware of the issue that the reviewer raised. It was suggested that vitamin A is required in the lung development (Timoneda J et al. Nutrients. 2018;10:1132), but evidence on the effects of vitamin A on lung size is limited. So, we are not sure how to explain about physiological perspectives regarding the inconsistent results for the FEV1/FVC ratio and COPD prevalence. On the basis of scatterplot results for the association between vitamin A levels and the FEV1/FVC ratio, some outliers of the FEV1/FVC ratio might influence the linear regression results leading to a lack of statistical significance. Because COPD is a binary variable, no outlier exists. We have now added this postulation in the discussion (page 9, revised).

  1. What is the normal range of hs CRP in a Korean population > 40 y. The current cut-off of 1 mg/L seems very low even in healthy individuals.

Response: Thank you for the reviewer's comments. The median and mean values of hs-CRP is 0.5 mg/L and 1 mg/L, respectively; 76% of the study participants showed 1 mg/L or less. Although the normal range of hs-CRP in Koreans has not been reported, hs-CRP levels < 1 mg/L was reported as the levels indicating a low risk of cardiovascular disease (Banait T et al, Cureus. 2022;14:e30225).

  1. In the methods the authors formulated three PF classifications. It seems that the restrictive group is overlooked in the results?

Response: The three PF classifications were formulated by the Korea Disease Control and Prevention Agency, which generated the national survey data that we used. When we analyzed associations for restrictive lung disease, which is a heterogeneous set of pulmonary disorders, as an outcome, we could not observe any significant association. So, we decided to focus on obstructive pulmonary disease for this manuscript.
